# WHY BE ADVERSARIAL? LET'S COOPERATE!: COOPERATIVE DATASET ALIGNMENT VIA JSD UPPER BOUND

## ABSTRACT

Unsupervised dataset alignment estimates a transformation that maps two or more source domains to a shared aligned domain given only the domain datasets. This task has many applications including generative modeling, unsupervised domain adaptation, and socially aware learning. Most prior works use adversarial learning (i.e., min-max optimization), which can be challenging to optimize and evaluate. A few recent works explore non-adversarial flow-based (i.e., invertible) approaches, but they lack a unified perspective. Therefore, we propose to unify and generalize previous flow-based approaches under a single non-adversarial framework, which we prove is equivalent to minimizing an upper bound on the Jensen-Shannon Divergence (JSD). Importantly, our problem reduces to a min-min, i.e., cooperative, problem and can provide a natural evaluation metric for unsupervised dataset alignment. We present empirical results of our framework on both simulated and real-world datasets to demonstrate the benefits of our approach.

## 1 INTRODUCTION

In many cases, a practitioner has access to multiple related but distinct datasets such as agricultural measurements from two farms, experimental data collected in different months, or sales data before and after a major event. *Unsupervised* dataset alignment (UDA) is the ML task aimed at aligning these related but distinct datasets in a shared space, which may be a latent space, *without* any pairing information between the two domains (i.e., unsupervised). This task has many applications such as generative modeling (e.g., (Zhu et al., 2017)), unsupervised domain adaptation (e.g., (Grover et al., 2020; Hu et al., 2018)), batch effect mitigation in biology (e.g., (Haghverdi et al., 2018)), and fairness-aware learning (e.g., (Zemel et al., 2013)).

The most common approach for obtaining such alignment transformations stems from Generative Adversarial Networks (GAN)(Goodfellow et al., 2014), which can be viewed as minimizing a *lower bound* on the Jensen-Shannon Divergence (JSD) between real and generated distributions. The lower bound is tight if and only if the inner maximization is solved perfectly. CycleGAN (Zhu et al., 2017) maps between *two* datasets via two GAN objectives between the two domains and a cycle consistency loss, which encourages approximate invertibility of the transformations. However, adversarial learning can be challenging to optimize in practice (see e.g. (Lucic et al., 2018; Kurach et al., 2019)) in part because of the competitive nature of the min-max optimization problem. Perhaps more importantly, the research community only has reasonable GAN evaluation metrics for certain data types. Specifically, the commonly accepted Frechet Inception Distance (FID) (Heusel et al., 2017) is only applicable to image or auditory data, which have standard powerful pretrained classifiers, and even the implementation of FID can have evaluation issues (Parmar et al., 2021). No clear metrics exist for tabular data or non-perceptual data.

Recently, flow-based methods that leverage invertible models have been proposed for the UDA task (Grover et al., 2020; Usman et al., 2020). AlignFlow (Grover et al., 2020) leverages invertible models to make the model cycle-consistent (i.e., invertible) *by construction* and introduce exact log-likelihood loss terms derived from standard flow-based generative models as a complementary loss terms to the adversarial loss terms. ~~Yet, AlignFlow still leverages adversarial learning.~~ Log-likelihood ratio minimizing flows (LRMF) (Usman et al., 2020) use invertible flow models and

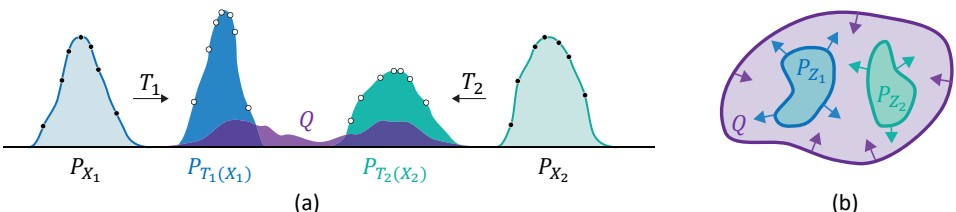

Figure 1: Overview of our proposed method. In our framework, domain specific transformation functions $\{T_j(\cdot)\}_{j=1}^k$ and a density model $Q$ are cooperatively trained to make the transformed representations be indistinguishable in the shared latent space. **(a)** 1-D example. By minimizing our proposed AUB loss, the transformation functions $T_1$ and $T_2$ are trained to map the corresponding distributions $P_{X_1}$ and $P_{X_2}$ to latent distributions $P_{T_1(X_1)}$ and $P_{T_2(X_2)}$ that have higher likelihood with respect to a base distribution $Q$. The density model $Q$, on the other hand, is trained to fit the mixture of the latent distributions $P_{T_1(X_1)}$ and $P_{T_2(x_2)}$. **(b)** Intuitively, the optimization process of our method can be seen to make the $Q$ distribution tight around the mixture of latent distributions to increase the likelihood (i.e., MLE) while the transformation functions $T_1$ and $T_2$ are encouraged to expand to fill the latent space defined by $Q$. Eventually, the latent distributions and $Q$ are converge to be the same distribution, which means that they are aligned.

density estimation for distribution alignment without adversarial learning and define a new metric based on the log-likelihood ratio. However, LRMF depends heavily on the density model class and can only partially align datasets if the target distribution is not in the chosen density model class. Additionally, the LRMF metric depends on this density model class and is only defined for two datasets.

Therefore, to provide an alternative to adversarial learning and generalize previous flow-based approaches, we propose a unified non-adversarial UDA framework, which we prove is equivalent to minimizing an upper bound on the JSD. Importantly, our problem reduces to a min-min, i.e., *cooperative*, problem, and the JSD upper bound can provide a natural evaluation metric for UDA that can be applied in any domain. Our framework requires two parts, the outer minimization requires an invertible model and the inner minimization requires a density model (e.g., Gaussian mixture models or normalizing flows (Dinh et al., 2017)). We summarize our contributions as follows:

- We prove that a minimization problem over density models is an *upper bound* on a generalized version of JSD that allows for more than two distributions. Importantly, we also theoretically quantify the bound gap and show that it can be made tight if the density model class is flexible enough.

- We use this JSD upper bound to derive a novel regularized loss function for UDA and explain its relationship to prior methods.

- We empirically demonstrate the benefits of our method compared to prior flow-based models on both simulated and real-world datasets.

**Notation** We will denote distributions as $P_X(\boldsymbol{x})$ where $X$ is the corresponding random variable. Invertible functions will be denoted by $T(\cdot)$. We will use $X_j \sim P_{X_j}$ to denote the observed random variable from the $j$-th distribution. We will use $Z_j \triangleq T_j(X_j) \sim P_{Z_j} \equiv P_{T_j(X_j)}$ to denote the latent random variable of the $j$-th distribution after applying $T_j$ to $X_j$ (and note that $X_j = T_j^{-1}(Z_j)$). We will denote the mixtures of these observed or latent distributions as $P_{X_{\text{mix}}} \triangleq \sum_j w_j P_{X_j}$ and $P_{Z_{\text{mix}}} \triangleq \sum_j w_j P_{Z_j}$, where $\boldsymbol{w}$ is a probability vector. We denote KL divergence, entropy, and cross entropy as $\text{KL}(\cdot, \cdot)$, $\text{H}(\cdot)$, and $\text{H}_c(\cdot, \cdot)$, respectively, where $\text{KL}(P, Q) = \text{H}_c(P, Q) - \text{H}(P)$.

## 2 REGULARIZED ALIGNMENT UPPER BOUND LOSS

In this section, we will introduce our main theoretical result proving an upper bound on the generalized JSD divergence, deriving our loss function based on this, and then showing that minimizing this upper bound results in aligned distributions assuming large enough capacity of the model components.

**Background: Normalizing Flows and Invertible Models** Normalizing flows are generative models that have tractable distributions where exact density evaluations and efficient samplings are ensured (Kobyzev et al., 2021). Such models leverage the change of variables formula to create an invertible mapping $T$ such that $P_X(\boldsymbol{x}) = P_Z(T(\boldsymbol{x}))|J_T(\boldsymbol{x})|$ where $P_Z$ is a known latent distribution and $|J_T(\boldsymbol{x})|$ is the absolute value of determinant of the Jacobian of the invertible map $T$. For sampling in distribution $P_X$, one need to first sample from the latent distribution $P_Z$ and then apply the inverse transform $T^{-1}$. Therefore the key challenge in designing invertible models is to have computationally efficient inverse evaluation and Jacobian determinant calculation. Many approaches have been proposed by parameterizing mapping function $T$ as deep neural networks including autoregressive structures (Kingma et al., 2016; Papamakarios et al., 2021), coupling layers (Dinh et al., 2017)(Kingma and Dhariwal, 2018), oridinary differential equations(Grathwohl et al., 2018), and invertible residual networks(Chen et al., 2019; Behrmann et al., 2019). Flow models can be then learned efficiently by maximizing the likelihood for the given data.

**Background: Generalized JSD** We remind the reader of the generalized Jensen-Shannon divergence for more than two distributions, where the standard JSD is recovered if $w_1 = w_2 = 0.5$.

**Definition 1** (Generalized Jensen-Shannon Divergence (GJSD) (Lin, 1991)). *Given $k$ distributions* $\{P_{X_j}\}_{j=1}^k$ *and a corresponding probability weight vector $\boldsymbol{w}$, the generalized Jensen-Shannon divergence is defined as (proof of equivalence in appendix):*

$$\begin{aligned} \text{GJSD}_{\boldsymbol{w}}(P_{X_1}, \cdots, P_{X_k}) &\triangleq \textstyle\sum_j w_j \, \text{KL}(P_{X_j}, \sum_j w_j P_{X_j}) \\ &\equiv \text{H}\left(\textstyle\sum_j w_j P_{X_j}\right) - \sum_j w_j \, \text{H}(P_{X_j}). \end{aligned} \quad (1)$$

## 2.1 GJSD UPPER BOUND

The goal of distribution alignment is to find a set of transformations $\{T_j(\cdot)\}_{j=1}^k$ (which will be invertible in our case) such that the latent distributions align, i.e., $P_{T_j(X_j)} = P_{T_{j'}(X_{j'})}$ or equivalently $P_{Z_j} = P_{Z_{j'}}$ for all $j \neq j'$. Given the properties of divergences, this alignment will happen if and only if $\text{GJSD}(P_{Z_1}, \cdots, P_{Z_k}) = 0$. Thus, ideally, we would minimize GJSD directly with respect to $T_j$, i.e.,

$$\min_{T_1, \cdots, T_k \in \mathcal{T}} \text{GJSD}(P_{T_1(X_1)}, \cdots, P_{T_k(X_k)}) \equiv \min_{T_1, \cdots, T_k \in \mathcal{T}} \text{H}\left(\textstyle\sum_j w_j P_{T_j(X_j)}\right) - \sum_j w_j \, \text{H}(P_{T_j(X_j)}),$$

where $\mathcal{T}$ is a class of invertible functions. However, we cannot evaluate the entropy terms in Eqn. 2 because we do not know the density of $P_{X_j}$; we only have samples from $P_{X_j}$. Therefore, we will upper bound the first entropy term in Eqn. 2 ($\text{H}\left(\sum_j w_j P_{X_j}\right)$) using an auxiliary density model and decompose the other entropy terms by leveraging the change of variables formula for invertible functions.

**Theorem 1** (GJSD Upper Bound). *Given an auxiliary density model class $\mathcal{Q}$, we form a GJSD upper bound:*

$$\text{GJSD}_{\boldsymbol{w}}(P_{Z_1}, \cdots, P_{Z_k}) \leq \min_{Q \in \mathcal{Q}} \text{H}_{\text{c}}(P_{Z_{\text{mix}}}, Q) - \textstyle\sum_j w_j \, \text{H}(P_{Z_j}),$$

*where the bound gap is exactly $\min_{Q \in \mathcal{Q}} \text{KL}(P_{Z_{\text{mix}}}, Q)$.*

*Proof of Theorem 1.* For any $Q \in \mathcal{Q}$, we have the following upper bound:

$$\begin{aligned} \text{GJSD}_{\boldsymbol{w}}(P_{Z_1}, \cdots, P_{Z_k}) &= \underbrace{\text{H}_{\text{c}}(P_{Z_{\text{mix}}}, Q) - \text{H}_{\text{c}}(P_{Z_{\text{mix}}}, Q)}_{=0} + \text{H}(P_{Z_{\text{mix}}}) - \textstyle\sum_j w_j \, \text{H}(P_{Z_j}) \\ &= \text{H}_{\text{c}}(P_{Z_{\text{mix}}}, Q) - \text{KL}(P_{Z_{\text{mix}}}, Q) - \textstyle\sum_j w_j \, \text{H}(P_{Z_j}) \\ &\leq \text{H}_{\text{c}}(P_{Z_{\text{mix}}}, Q) - \textstyle\sum_j w_j \, \text{H}(P_{Z_j}), \end{aligned}$$

where the inequality is by the fact that KL divergence is non-negative and the bound gap is equal to $\text{KL}(P_{Z_{\text{mix}}}, Q)$. The $Q$ that achieves the minimum in the upper bound is equivalent to the $Q$ that

minimizes the bound gap, i.e.,

$$Q^* = \underset{Q \in \mathcal{Q}}{\arg\min} \ \mathrm{H_c}(P_{Z_\mathrm{mix}}, Q) \underbrace{- \sum_j w_j \, \mathrm{H}(P_{Z_j})}_{\text{Constant w.r.t. } Q} \tag{2}$$

$$= \underset{Q \in \mathcal{Q}}{\arg\min} \ \mathrm{H_c}(P_{Z_\mathrm{mix}}, Q) \underbrace{- \mathrm{H}(P_{Z_\mathrm{mix}})}_{\text{Constant w.r.t. } Q} \tag{3}$$

$$= \underset{Q \in \mathcal{Q}}{\arg\min} \ \mathrm{KL}(P_{Z_\mathrm{mix}}, Q) . \tag{4}$$

$$\square$$

The tightness of the bound depends on how well the class of density models $\mathcal{Q}$ (e.g., mixture models, normalizing flows, or autoregressive densities) can approximate $P_{Z_\mathrm{mix}}$; notably, the bound can be made tight if $P_{Z_\mathrm{mix}} \in \mathcal{Q}$. Also, one key feature of this upper bound is that the cross entropy term can be evaluated using only samples from $P_{X_j}$ and the transformations $T_j$, i.e., $\mathrm{H_c}(P_{Z_\mathrm{mix}}, Q) = \sum_j w_j \mathbb{E}_{P_{X_j}}[-\log Q(T_j(\boldsymbol{x}_j))]$. However, we still cannot evaluate the other entropy terms $\mathrm{H}(P_{Z_j})$ since we do not know the densities of $P_{Z_j}$ (or $P_{X_j}$). Thus, we leverage the fact that the $T_j$ functions are invertible to define an entropy change of variables.

**Lemma 2** (Entropy Change of Variables). *Let $X \sim P_X$ and $Z \triangleq T(X) \sim P_Z$, where $T$ is an invertible transformation. The entropy of $Z$ can be decomposed as follows:*

$$\mathrm{H}(P_Z) = \mathrm{H}(P_X) + \mathbb{E}_{P_X}[\log |J_T(\boldsymbol{x})|] , \tag{5}$$

*where $|J_T(\boldsymbol{x})|$ is the determinant of the Jacobian of $T$.*

The key insight from this lemma is that $\mathrm{H}(P_X)$ is a constant with respect to $T$ and can thus be ignored when optimizing $T$, while $\mathbb{E}_{P_X}[\log |J_T(\boldsymbol{x})|]$ can be approximated using only samples from $P_X$.

## 2.2 ALIGNMENT UPPER BOUND (AUB)

Combining Theorem 1 and Lemma 2, we can arrive at our final objective function which is equivalent to minimizing an upper bound on the GJSD:

$$\mathrm{GJSD}_{\boldsymbol{w}}(P_{Z_1}, \cdots, P_{Z_k}) \leq \min_{Q \in \mathcal{Q}} \mathrm{H_c}(P_{Z_\mathrm{mix}}, Q) - \sum_j w_j \, \mathrm{H}(P_{Z_j}) \tag{6}$$

$$= \min_{Q \in \mathcal{Q}} \sum_j w_j \mathbb{E}_{P_{X_j}}[-\log Q(T_j(\boldsymbol{x}))|J_{T_j}(\boldsymbol{x})|] - \sum_j w_j \, \mathrm{H}(P_{X_j}) , \tag{7}$$

where the last term $-\sum_j w_j \, \mathrm{H}(P_{X_j})$ is constant with respect to $T_j$ functions so they can be ignored. We formally define this loss function as follows.

**Definition 2** (Alignment Upper Bound Loss). *Given $k$ continuous distributions $\{P_{X_j}\}_{j=1}^k$, a class of continuous distributions $\mathcal{Q}$, and a probability weight vector $\boldsymbol{w}$, the alignment upper bound loss is defined as follows:*

$$\mathcal{L}_{\mathrm{AUB}}(T_1, \cdots, T_k; \{P_{X_j}\}_{j=1}^k, \mathcal{Q}, \boldsymbol{w}) \triangleq \min_{Q \in \mathcal{Q}} \sum_j w_j \mathbb{E}_{P_{X_j}}[-\log |J_{T_j}(\boldsymbol{x})| \, Q(T_j(\boldsymbol{x}))] , \tag{8}$$

*where $T_j$ are invertible and $|J_{T_j}(\boldsymbol{x})|$ is the absolute value of the Jacobian determinant.*

Notice that this alignment loss can be seen as learning the best base distribution given fixed flow models $T_j$. We now consider the theoretical optimum if we optimize over all invertible functions.

**Theorem 3** (Alignment at Global Minimum of $\mathcal{L}_{\mathrm{AUB}}$). *If $\mathcal{L}_{\mathrm{AUB}}$ is minimized over the class of all invertible functions, a global minimum of $\mathcal{L}_{\mathrm{AUB}}$ implies that the latent distributions are aligned, i.e., $P_{T_j(X_j)} = P_{T_{j'}(X_{j'})}$ for all $j \neq j'$. Notably, this result holds regardless of $\mathcal{Q}$.*

Informally, this can be proved by showing that the problem decouples into separate normalizing flow losses where $Q$ is the base distribution and the optimum is achieved only if $P_{T_j(X_j)} = Q$ for all $T_j$ (formal proof in the appendix). This alignment of the latent distributions also implies the translation between any of the *observed* component distributions. The proof follows directly from Theorem 3 and the change of variables formula.

**Corollary 4** (Translation at Global Minimum of $\mathcal{L}_{\mathrm{AUB}}$). *Similar to Theorem 3, a global minimum of $\mathcal{L}_{\mathrm{AUB}}$ implies translation between any component distributions using the inverses of $T_j$, i.e., $P_{T_{j'}^{-1}(T_j(X_j))} = P_{X_{j'}}$ for all $j \neq j'$.*

**Regularization via Transportation Cost**   While the alignment objective is the most challenging part of UDA, we suggest that regularization may also be useful for practical and stable alignment (or translation) between datasets because there are many optimal alignment solutions—even infinitely many in most cases (see appendix for two examples). We alleviate this issue by adding expected transportation cost (usually squared Euclidean distance) as a regularization to our objective inspired by optimal transport (OT) concepts.

**Definition 3** (Regularized Alignment Upper Bound Loss, RAUB). *Given similar setup as in Def. 2 and a transportation cost function $c(a, b) \geq 0$ for transporting a point from $a$ to $b$, the regularized alignment upper bound loss is defined as:*

$$
\begin{aligned}
&\mathcal{L}_{\text{RAUB}}(T_1, \cdots, T_k; \{P_{X_j}\}_{j=1}^k, \mathcal{Q}, \boldsymbol{w}, \lambda, c) \\
&\triangleq \min_{Q \in \mathcal{Q}} \sum_j w_j \mathbb{E}_{P_{X_j}} \left[ -\log |J_{T_j}(\boldsymbol{x})| \, Q(T_j(\boldsymbol{x})) + \lambda c(\boldsymbol{x}, T_j(\boldsymbol{x})) \right].
\end{aligned}
\tag{9}
$$

Pseudo-code for training our model with RAUB objective can be found in the appendix Section C.

## 3   RELATIONSHIP TO PRIOR WORKS

**AlignFlow without adversarial terms is a special case**   AlignFlow (Grover et al., 2020) *without* adversarial loss terms is a special case of our method for two distributions where the density model class $\mathcal{Q}$ only contains the standard normal distribution (i.e., a singleton class) and no regularization is used (i.e., $\lambda = 0$). Thus, AlignFlow can be viewed as initially optimizing a poor upper bound on JSD; however, the JSD bound becomes tighter as training progresses because the latent distributions *independently* move towards the same normal distribution.

**LRMF is special case with only one transformation**   Log-likelihood ratio minimizing flows (LRMF) (Usman et al., 2020) is also a special case of our method for only two distributions, where one transformation is fixed at the identity (i.e., $T_2 = \text{Id}$) and no regularization is applied (i.e., $\lambda = 0$). While the final practical LRMF objective is a special case of ours, the theory is developed from a different but complementary perspective. The LRMF metric developed requires an assumption about a given density model class, which enables a zero point (or absolute value) of the metric to be estimated but requires fitting extra domain density models. Usman et al. (2020) also do not uncover the connection of the objective as an upper bound on JSD regardless of the density model class. Additionally, to ensure alignment, LRMF requires that the density model class includes the true target distribution because only one invertible transform is used, while our approach can theoretically align even if the shared density model class is weak (see Theorem 3 and our simulated experiments). See Fig. 2 for a comparison of our approach to prior works.

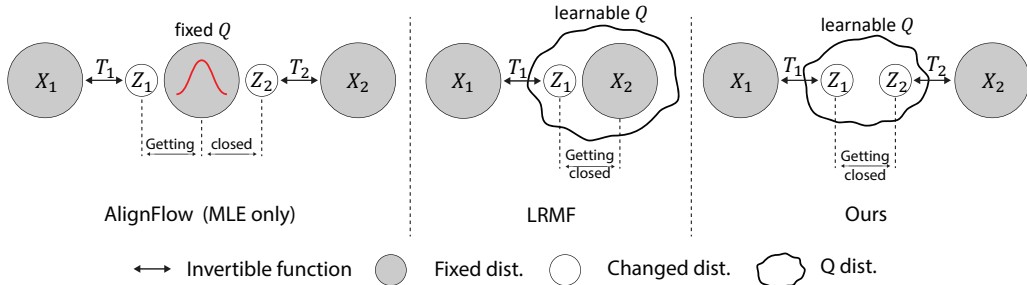

Figure 2: High-level comparison with the baseline models. AlignFlow, LRMF, and our setup are illustrated in a row from left. Transformation functions in AlignFlow are independently trained to be fitted to the fixed standard gaussian distribution. $T_1$ in LRMF is aimed to directly map the given $X_1$ to another image distribution $X_2$. The density model $Q$ in LRMF is not fixed and learned to fit to $\{Z_1 \cap X_2\}$. In our setup, $T_1$ and $T_2$ are trained to obtain the high likelihood from the learnable $Q$ distribution which is fitted to the shared latent distributions. In every setup, the latent distributions $Z_1$ and $Z_2$ are getting closer to the target distribution as training goes by. Details are provided in Section 3.

**Cooperative networks versus adversarial networks** Analogous to the generator $G$ and the discriminator $D$ in adversarial learning, our framework has two main networks, $T_j$ and $Q_z$. We can use *any* invertible function for $T_j$ (e.g., coupling-based flows (Dinh et al., 2017), neural ODE flows (Grathwohl et al., 2018), or residual flows (Chen et al., 2019)) and *any* (approximate) density models for $Q_z$ (e.g., kernel densities (in low dimensions), mixture models, autoregressive densities (Salimans et al., 2017), normalizing flows (Kingma and Dhariwal, 2018), or even VAEs (Kingma and Welling, 2019)). Thus, our framework has similar modularity compared to adversarial approaches. In contrast, we have a min-min, i.e., cooperative, optimization problem, but our transformations must be invertible. As another difference, the inner optimization problem may be more challenging (i.e., fitting a density model $Q_z$ may be more difficult than fitting an auxiliary discriminator $D$), but the overall min-min is likely to be more stable than the min-max problem—specifically, note that our problem can align even if the density model $Q_z$ is not optimal (see Theorem 3). We expect some of these limitations to be alleviated as new invertible models and density models are continually being developed. Therefore, the proposed approach provides a fundamental alternative to adversarial learning with different strengths and weaknesses.

## 4 EXPERIMENT

### 4.1 2D DATASET COMPARISON WITH RELATED WORKS

We compare our method with related works in order to illustrate the limitations of prior methods and how our method overcomes these limitations. All implementation details for the 2D dataset are available in the appendix.

**Single $T$ vs. Double $T$ (LRMF vs. Ours)** We first compare our method with LRMF (Usman et al., 2020) method. We construct the experiment to have the task: Translation between the two half-circled distributions: $X_1$ and $X_2$ in the moons dataset. In this example, we made two models, one with LRMF setup and one with our AUB setup. As illustrated in Fig. 3, the LRMF method fails to transform between $X_1$ and $X_2$. Even though $Q$ can model well enough for $T_1(X_1)$, $Q$ can only model the mean and variance of $X_2$ which is obviously not informative enough. Therefore, the LRMF fails to transform between two datasets. While in the AUB setup, both $T_1(X_1)$ and $T_2(X_2)$ are modeled to the same distribution which $Q$ can be learned to fit with high likelihood which leads to better translation results. In conclusion, the performance of the LRMF model is limited by the power of the density model $Q$ which means if $Q$ fails to model one of the domain distribution in high likelihood, data alignment cannot be achieved with good performances.

**Simple Fixed $Q$ vs. Learnable $Q$ (AlignFlow vs. Ours)** Next we compare our method with AlignFlow(Grover et al., 2020; Hu et al., 2018) setup. We construct the experiment to have the task: Transform between the two random patterns $X_1$ and $X_2$ from the randomly generated datasets. Again, we made two models with AlignFlow and our AUB setups respectively. As illustrated in Fig. 3, the AlignFlow method fails to transform between $X_1$ and $X_2$, because the transformed dataset $T_1(X_1)$ and $T_2(X_2)$ failed to reach the normal distribution $Q$. While in the AUB setup, the density model $Q$ is learned to help fit the transformed distributions $T_1(X_1)$ and $T_2(X_2)$, which allows them to be aligned with each other easier. In conclusion, the performance of the AlignFlow model is limited by the performances of the invertible functions.

**Regularized vs. Un-regularized (Some prior works vs. Ours)** We finally show the importance of the regularization term. We construct the experiment to have the task: Transform between two concentric circles with the same mean but slighted different radius. In this example, we make two models with our AUB approach, but one with regularization and one without. As illustrated in the Figure 4, both models are able to transform between two distributions. However, the transformation pattern is not natural in terms of the movement of each point. Each pair created by the unregularized model has bigger transportation cost compared to the pairs created by the regularized model. Therefore, we suggest that by adding transportation cost, the resulting transformation between samples will be closer to the identity function and therefore more stable.

### 4.2 REAL-WORLD DATASETS

**Metrics** In this subsection, we use two metrics to measure the overall model performances: Frechet Inception Distance (FID) (Heusel et al., 2017) implemented by (Seitzer, 2020) and our metric Alignment Upper Bound (AUB).

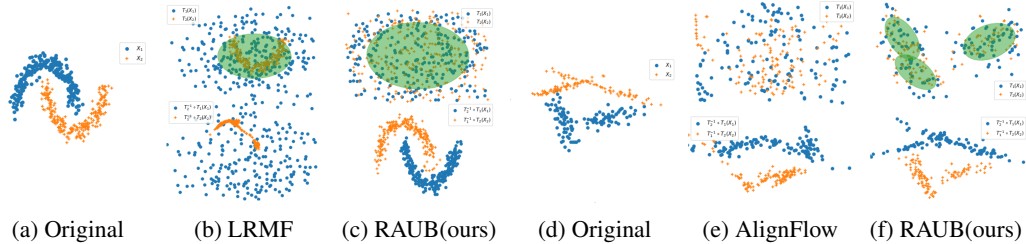

(a) Original      (b) LRMF      (c) RAUB(ours)      (d) Original      (e) AlignFlow      (f) RAUB(ours)

Figure 3: Top row is latent space and bottom is the data translated into the other space. (a-c) LRMF, which only has *one* transformation $T$ may not be able to align the datasets if the density model class $\mathcal{Q}$ is not expressive enough (in this case Gaussian distributions) while using two transformations as in our framework can align them. (d-f) AlignFlow (without adversarial terms) may not align because $Q_z$ is fixed at a standard normal, while our approach with learnable mixture of Gaussians for $Q_z$ is able to learn an alignment (both use the same $T_j$ models).

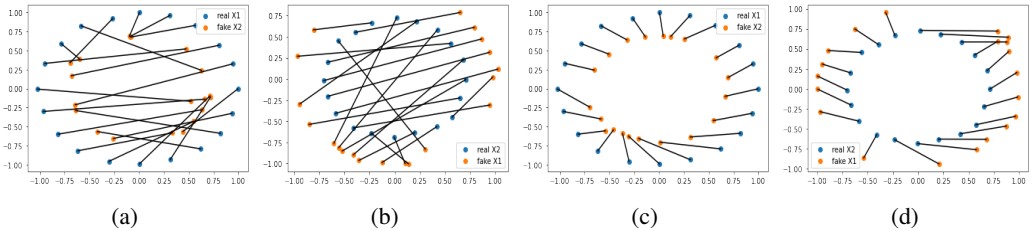

(a)          (b)          (c)          (d)

Figure 4: An unregularized alignment loss (figure (a) and (b)) can lead to excessive and unexpected movement of points in the latent representation (lines connect transported points), while our regularized alignment loss ((figure (c) and (d))) yields a unique and regularized solution that moves points significantly less and is closer to the identity function.

**Unsupervised domain translation** We first perform three image translation tasks on MNIST dataset; more specifically, we are flipping digit images in "0/1", "0/2" , and "1/2" pairs shown in Fig. 5. We use RealNVP invertible models for all translation maps $T_j$, as well as the density model $Q$. We evaluate our model performances in Table 1 along with baseline models: LRMF and MLE version of AlignFlow. Note that all methods are flow-based models and thus images translated back to original domain are exactly the same, which implies exact cycle consistency.

As represented in Table. 1, both of our approaches outperform the baseline models in terms of FID and AUB. The lower score in AUB indicates our model has the tighter bound than baseline model. This implies the statistical distance between the shared distribution $\{Z_j\}_{j=1}^{k}$ and our density model $Q$ is smaller than baselines, meaning our model has the better dataset alignment performance. We believe this result comes from our model setup, i.e., a learnable density model and domain specific invertible transformation functions with shared space. Specifically, AlignFlow with a fixed standard normal distribution as their $Q$ obtains worse AUB because the $Q$ is not powerful enough to model the complex shared space trained from the real world dataset. On the other hand, LRMF shows the lack of stability when trained with relatively simple models that we are using, i.e., RealNVP $T$ and Real NVP $Q$. This is because the transformation function and the density model should be able to directly model the complex real distribution in their setup. By comparing our approach with and without transportation cost, we observe introducing transportation cost yields insignificant decrease in AUB and an increase in FID. This indicates the transportation cost regularizes our method well without degeneration, so that our method can find better solution (which is theoretically unique).

The impressive performance of our method in AUB shows a consistent pattern in FID score as well. Across entire translation cases, our method shows the better performance in terms of FID. This result can be intuitively understood by Fig. 5. AlignFlow shows less stable translation results than our method especially for translating to digit '2' from digits '0' and '1'. We believe this phenomenon comes from the lack of expressivity of their $Q$ model. On the other hand, our model shows stable results across all translation cases, which are also quantitatively verified via the lower FID score. Also, from the comparison on the latent representations (second column of each macro column) of

ours with and without transportation cost, we can observe that the latent representation of the $\lambda = 1$ setup contains numerous excessive values. This implies adding transportation cost ($lambda = 1$) prevents the excessive movement in the latent space, thus, our model can find a simpler and smoother solution during the training.

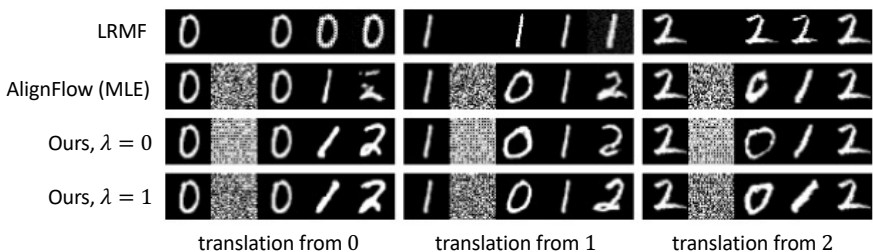

translation from 0  translation from 1  translation from 2

Figure 5: All pair-wise translation results among MNIST digits 0-2 where each block has first image as the original digit followed by the latent image, and three corresponding translated digits. Please note that second column of LRMF is set to be black because it does not have latent representation, and LRMF fails to translate in this situation which is why the numbers are all the same.

Table 1: FID and AUB score for three images translation tasks in MNIST (for both metrics, lower the better). FID score for each translation task is calculated by averaging scores from each direction and AUB score is shown in nats. This table shows that our model has overall better performances than all baselines models in terms of both metrics.

| | FID | | | | AUB | | | |
|---|---|---|---|---|---|---|---|---|
| | $0\leftrightarrow1$ | $0\leftrightarrow2$ | $1\leftrightarrow2$ | Avg. | $0\leftrightarrow1$ | $0\leftrightarrow2$ | $1\leftrightarrow2$ | Avg. |
| AlignFlow (MLE) | 38.90 | 71.17 | 61.33 | 57.13 | -4797.27 | -4504.20 | -4834.30 | -4711.92 |
| LRMF | 224.02 | 141.70 | 182.31 | 182.68 | -713.54 | -592.90 | -1323.75 | -876.73 |
| Ours, $\lambda = 0$ | 31.26 | **43.29** | 41.55 | 38.70 | **-4824.69** | **-4555.30** | **-4862.84** | **-4747.61** |
| Ours, $\lambda = 1$ | **29.30** | 45.21 | **39.95** | **38.15** | -4819.73 | -4547.76 | -4857.67 | -4741.72 |

**Multi-domain translation** To illustrate that our method can be easily scaled to more domain distributions, we present qualitative examples of translating between every digit and every other digit for MNIST in Fig. 6 with performances in Table 2. To show the effect of transportation cost proposed in our paper, we also compare our model with and without transportation cost. Note that we omit LRMF in this experiment because the multi-domain situation is hard to deal with for LRMF setup due to LRMF's two distribution setup and assymetric model structure.

As shown in Table 2, our approach with and without transportation cost shows better performance in terms of FID and AUB than AlignFlow since our learnable density estimator can model more complex distribution than fixed simple density model in AlignFlow. In other words, the shared space of AlignFlow is limited because of the fixed simple density model. Note that the gap in the AUB score between AlignFow and our model is larger in this multi-domain setting (Table 2) than in the two domain case (Table 1). We hypothesize that this is because the shared density model across many domains needs to be more complex than for only two domains and thus, our learnable $Q$ can create a better upper bound to optimize than AlignFlow. In this experiment, while ours with transportation cost performs slightly worse, the results are comparable and it does provide a natural regularization that ensures a theoretically identifiable solution.

The superiority of our method compared to AlignFlow in multiple-domain translation can also be verifed through qualitative comparisons in Fig. 6. The leftmost column is an input image and the second and third macro columns are the results from ours and AlignFlow. by forwarding a given $k$-th latent $T_k(x_k)$ into 10 inverse transformation functions, respectively. It is easy to observe that ours has more clear results in most cases than baseline. Moreover, our model shows better performance in maintaining the original identity (e.g., width and type of a stroke) than the baseline, as seen in third, fourth and ninth rows. This is because we jointly train our transformation functions with a learnable density model, while AlignFlow independently train their transformation functions. This benefit of our approach may be crucial for other datasets such as human faces (Choi et al., 2018) where maintaining the original identity is important.

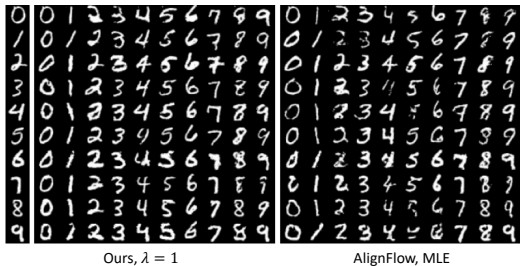

Figure 6: Qualitative comparison on translation results across 10 classes with AlignFlow and ours with transportation cost.

Table 2: FID and AUB score for domain alignment task in 10 domains. FID score is calculated by average across all paired translations and AUB score is shown in nats

|  | FID | AUB |
|---|---|---|
| AlignFlow (MLE) | 49.82 | -4661.05 |
| Ours, $\lambda = 0$ | 43.25 | -4715.02 |
| Ours, $\lambda = 1$ | 44.06 | -4697.42 |

## 5 DISCUSSION AND CONCLUSION

**Transfer Learning Capability**   The performance of our model can be improved by adapting pre-trained density models. Fig. 7(a) shows preliminary results. Specifically, a MLE version of Align-Flow model may still struggle to translate between USPS and MNIST dataset after learning for 200 epochs while ours can show qualitative results after only 1 epoch if we are using pretrained powerful density model for $Q$ Salimans et al. (2017).

**Generative Tasks**   Our model is also capable of generating samples in each domain. One needs to sample from the density model $Q$ to have a latent image $z$ first, and then forward to the inverse function $T_j^{-1}$ to get the image sampled in $j^{th}$ domain. Examples of generated images for each domain in MNIST data are shown in Fig. 7(b). The quality of the generative result also reflects the tightness of the bound between the latent space and the latent density model as illustrated in Eqn. 4.

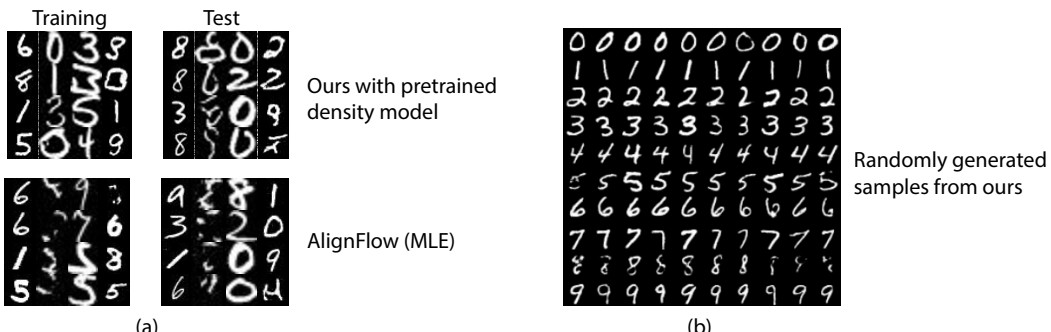

Figure 7: Figure (a) is translation result between USPS and MNIST dataset. Top two images are the training and testing result if using pretrained PixelCNN++(Salimans et al., 2017) models after only 1 epochs, and the bottom images are the results of MLE version of AlignFlow models after 200 epochs. Figure (b) shows the generated images of our model for all domains. Model used in this task is adapted from multi-domain translation tasks in subsection 4.2.

In this paper, we proposed a novel upper bound on the generalized JSD that leads to a theoretically grounded alignment loss function. We then show that this framework unifies previous flow-based approaches to dataset alignment and demonstrate the benefits of our approach compared to prior flow-based methods. Additionally, we show that our framework could enable the use of pre-trained density models which would enable a type of transfer learning for distribution alignment. More broadly, we expect that our AUB metric can be useful as a domain-agnostic metric for comparing dataset alignment methods beyond images. An alignment metric that is not tied to a particular pretrained model (as for FID) or to a particular data type will be critical for systematic progress in unsupervised dataset alignment. We hope this paper provides one step in that direction.

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

## A    PROOFS

*Proof of Equivalence in Def. 1.* While the proof of the equivalence is well-known, we reproduce here for completeness. As a reminder, the KL divergence is defined as:

$$\mathrm{KL}(P, Q) = \mathbb{E}_P[\log \tfrac{P(x)}{Q(x)}] = \mathbb{E}_P[-\log Q(x)] - \mathbb{E}_P[-\log P(x)] = \mathrm{H_c}(P, Q) - \mathrm{H}(P)\,, \quad (10)$$

where $\mathrm{H_c}(\cdot, \cdot)$ denotes the cross entropy and $\mathrm{H}(\cdot)$ denotes entropy. Given this, we can now easily derive the equivalence:

$$\mathrm{GJSD}_{\boldsymbol{w}}(P_{X_1}, \cdots, P_{X_k}) = \textstyle\sum_j w_j \, \mathrm{KL}(P_{X_j}, P_{X_{\mathrm{mix}}}) \tag{11}$$

$$= \textstyle\sum_j w_j (\mathrm{H_c}(P_{X_j}, P_{X_{\mathrm{mix}}}) - \mathrm{H}(P_{X_j})) \tag{12}$$

$$= \textstyle\sum_j w_j \, \mathrm{H_c}(P_{X_j}, P_{X_{\mathrm{mix}}}) - \sum_j w_j \, \mathrm{H}(P_{X_j}) \tag{13}$$

$$= \textstyle\sum_j w_j \mathbb{E}_{P_{X_j}}[-\log P_{X_{\mathrm{mix}}}] - \sum_j w_j \, \mathrm{H}(P_{X_j}) \tag{14}$$

$$= \textstyle\sum_j w_j \int_{\mathcal{X}} -P_{X_j}(x) \log P_{X_{\mathrm{mix}}}(x) dx - \sum_j w_j \, \mathrm{H}(P_{X_j}) \tag{15}$$

$$= \int_{\mathcal{X}} -\textstyle\sum_j w_j P_{X_j}(x) \log P_{X_{\mathrm{mix}}}(x) dx - \sum_j w_j \, \mathrm{H}(P_{X_j}) \tag{16}$$

$$= \int_{\mathcal{X}} -P_{X_{\mathrm{mix}}}(x) \log P_{X_{\mathrm{mix}}}(x) dx - \textstyle\sum_j w_j \, \mathrm{H}(P_{X_j}) \tag{17}$$

$$= \mathrm{H}(P_{X_{\mathrm{mix}}}) - \textstyle\sum_j w_j \, \mathrm{H}(P_{X_j})\,. \tag{18}$$

□

*Proof of Lemma 2.* First, we note the following fact from the standard change of variables formula:

$$P_X(\boldsymbol{x}) = P_Z(T(\boldsymbol{x}))|J_T(\boldsymbol{x})|$$
$$\Rightarrow P_X(\boldsymbol{x})|J_T(\boldsymbol{x})|^{-1} = P_Z(T(\boldsymbol{x}))\,. \tag{19}$$

We can now derive our result using the change of variables for expectations (i.e., LOTUS) and the probability change of variables from above:

$$\mathrm{H}(P_Z) = \mathbb{E}_{P_Z}[-\log P_Z(\boldsymbol{z})] = \mathbb{E}_{P_X}[-\log P_Z(T(\boldsymbol{x}))]$$
$$= \mathbb{E}_{P_X}[-\log P_X(\boldsymbol{x})|J_T(\boldsymbol{x})|^{-1}]$$
$$= \mathbb{E}_{P_X}[-\log P_X(\boldsymbol{x})] + \mathbb{E}_{P_X}[-\log |J_T(\boldsymbol{x})|^{-1}]$$
$$= \mathrm{H}(P_X) + \mathbb{E}_{P_X}[\log |J_T(\boldsymbol{x})|]\,.$$

□

*Proof of Theorem 3.* Given any fixed $Q$, minimizing $\mathcal{L}_{\mathrm{AUB}}$ decouples into minimizing separate normalizing flow losses where $Q$ is the base distribution. For each normalizing flow, there exists an invertible $T_j$ such that $T_j(X_j) \sim Q$, and this achieves the minimum value of $\mathcal{L}_{\mathrm{AUB}}$. More formally,

$$\min_{T_1, \cdots, T_k} \mathcal{L}_{\mathrm{AUB}}(T_1, \cdots, T_k) \tag{20}$$

$$= \min_{T_1, \cdots, T_k} \textstyle\sum_j w_j \mathbb{E}_{P_{X_j}}[-\log |J_{T_j}(\boldsymbol{x})| \, Q(T_j(\boldsymbol{x}))] \tag{21}$$

$$= \textstyle\sum_j w_j \min_{T_j} \mathbb{E}_{P_{X_j}}[-\log |J_{T_j}(\boldsymbol{x})| \, Q(T_j(\boldsymbol{x}))] + \mathrm{H}(P_{X_j}) - \mathrm{H}(P_{X_j}) \tag{22}$$

$$= \textstyle\sum_j w_j \min_{T_j} \mathbb{E}_{P_{X_j}}[-\log |J_{T_j}(\boldsymbol{x})| \, Q(T_j(\boldsymbol{x}))] + \mathrm{H}(P_{X_j}) - \mathbb{E}_{P_{X_j}}[-\log P_{X_j}(\boldsymbol{x})]) \tag{23}$$

$$= \textstyle\sum_j w_j \, \mathrm{H}(P_{X_j}) + \sum_j w_j \min_{T_j} \mathbb{E}_{P_{X_j}}[\log \tfrac{P_{X_j}(\boldsymbol{x})|J_{T_j}(\boldsymbol{x})|^{-1}}{Q(T_j(\boldsymbol{x}))}] \tag{24}$$

$$= \textstyle\sum_j w_j \, \mathrm{H}(P_{X_j}) + \sum_j w_j \min_{T_j} \mathbb{E}_{P_{X_j}}[\log \tfrac{P_{T_j(X_j)}(T_j(\boldsymbol{x}))}{Q(T_j(\boldsymbol{x}))}] \tag{25}$$

$$= \textstyle\sum_j w_j \, \mathrm{H}(P_{X_j}) + \sum_j w_j \min_{T_j} \mathbb{E}_{P_{T_j(X_j)}}[\log \tfrac{P_{T_j(X_j)}(\boldsymbol{z})}{Q(\boldsymbol{z})}] \tag{26}$$

$$= \textstyle\sum_j w_j \, \mathrm{H}(P_{X_j}) + \sum_j w_j \min_{T_j} \mathrm{KL}(P_{T_j(X_j)}, Q)\,. \tag{27}$$

Given that $\mathrm{KL}(P, Q) \geq 0$ and equal to 0 if and only if $P = Q$, the global minimum is achieved only if $P_{T_j(X_j)} = Q, \forall j$ and there exist such invertible functions (e.g., the optimal Monge map between $P_{X_j}$ and $Q$ for squared Euclidean transportation cost (Peyré and Cuturi, 2019)). Additionally, the optimal value is $\sum_j w_j \, \mathrm{H}(P_{X_j})$, which is constant with respect to the $T_j$ transformations.     □

## B    EXAMPLES OF NON-UNIQUE ALIGNMENT SOLUTIONS

### B.1    GAUSSIAN EXAMPLE

Suppose the component distributions are normal distributions, i.e., $X_1 \sim \mathcal{N}(\mu_1, I)$ and $X_2 \sim \mathcal{N}(\mu_2, I)$, and for even greater simplicity, we assume $T_2$ is the identity, i.e., $T_2(\boldsymbol{x}) = \boldsymbol{x}$. Then, a global optimal solution could be $T_1(\boldsymbol{x}) = U(\boldsymbol{x} - \mu_1 + \mu_2)$ for *any* orthogonal matrix $U$, i.e., there are infinitely many invertible functions that align the distributions. Note that this lack of unique solutions is not restricted to orthogonal rotations (see appendix for a more complex example).

### B.2    COMPLEX EXAMPLE

Consider the 1D case where $\mathcal{Q}$ only contains the uniform distribution. Thus, $T_1$ and $T_2$ must map their distributions to the uniform distribution for alignment. One solution would be that $T_1 = F_1$ and $T_2 = F_2$ where $F_1$ and $F_2$ are the CDFs of $P_{X_1}$ and $P_{X_2}$. Yet, there are infinitely many other possible solutions. Consider an invertible function that subdivides the unit interval into an arbitrarily large number of equal length intervals and then shuffles these intervals with a fixed arbitrary permutation. More formally, we could define this as:

$$
S_{m,\pi}(x) = \begin{cases} x - \frac{1}{m} + \frac{\pi(1)}{m} & \text{if } x \in [0, \frac{1}{m}) \\ x - \frac{2}{m} + \frac{\pi(2)}{m} & \text{if } x \in [\frac{1}{m}, \frac{2}{m}) \\ \vdots & \vdots \\ x - \frac{m}{m} + \frac{\pi(m)}{m} & \text{if } x \in [\frac{m-1}{m}, 1] \end{cases} \tag{28}
$$

where $\pi(\cdot)$ is a permutation of the integers 1 to $m$. Given this, then other optimal solutions could be $T_1 = S_{m,\pi} \circ F_1$ and $T_2 = F_2$ for any $m > 1$ and any permutation $\pi$. This idea could be generalized to higher dimensions as well by mapping to the multivariate uniform distribution and subdividing the unit hypercube similarly.

## C    ALGORITHM

We summarize our computation of regularized alignment upper bound in Alg. 1.

---

**Algorithm 1** Training algorithm for our model

---

**Input:** datasets $\{X_j\}_{j=1}^k$ for $k$ domains; normalizing flows $\{T_j(x_j; \theta_j)\}_{j=1}^k$; density model $Q(z; \phi)$; learning rate $\eta$; transportation cost factor $\lambda$ maximum epoch $E_{max}$; initial parameters value $\{T_j^{(0)}(x_j; \theta_j)\}_{j=1}^k, Q^{(0)}(z; \phi)$;
**Output:** $\{\theta^*\}_j^k$;
   **for** epoch $= 1, E_{max}$ **do**
      **for** each batch $\{x_j\}_{j=1}^k$ **do**
         $\phi \leftarrow \phi + \eta \cdot \lambda_\theta \log Q(\{T_j(x_j; \phi)\}_{j=1}^k)$         $\triangleright$ Update $Q$ using data from all domains
      **end for**
      **for** each batch $\{x_j\}_{j=1}^k$ **do**
         **for** domain$j = 1, k$ **do**
            $\theta \leftarrow \theta + \eta \cdot \log |J_{T_j}(\boldsymbol{x})| Q(T_j(\boldsymbol{x})) - \lambda c(\boldsymbol{x}, T_j(\boldsymbol{x}))$ $\triangleright$ Update $T$ independently across all domains
         **end for**
      **end for**
   **end for**

---

## D    DETAILED PARAMETERS USED IN 2D DATASET EXPERIMENT

### D.1    LRMF VS. OURS EXPERIMENT

- $T$ for LRMF setup: $T_1$: 8 channel-wise mask for Real-NVP model with $s$ and $t$ derived from 64 hidden channels of fully connected networks. $T_2$: Identity function.

- $T$ for RAUB setup: $T_1$ and $T_1$: 8 channel-wise mask for Real-NVP model with $s$ and $t$ derived from 64 hidden channels of fully connected networks. Regularization coefficient $\lambda = 0$

- $Q$ for both: A single Gaussian distribution with trainable mean and trainable variances.

## D.2 ALIGNFLOW VS. OURS EXPERIMENT

- $T$ for both: 2 channel-wise mask for RealNVP model with $s$ and $t$ derived from 8 hidden channels of fully connected networks.

- $Q$ for Alignflow setup: A single fixed normal distribution.

- $Q$ for RAUB setup: A learnable mixture of Gaussian with 3 components. Regularization coefficient $\lambda = 0$

## D.3 REGULARIZED VS. UNREGULARIZED EXPERIMENT

1. $T$ for both: 8 channel-wise mask for RealNVP model with $s$ and $t$ derived from 64 hidden channels of fully connected networks.

2. $Q$ for both: A learnable mixture of Gaussian with 2 components.

3. $\lambda$ for unregularized Experiment : $\lambda = 0$

4. $\lambda$ for regularized Experiment : $\lambda = 1$

## E INTERPOLATION OVER LATENT SPACE

We use the same model used in the "Multi-domain translation" part in section 4.2 and perform interpolation in the latent space. We first randomly select two distinct real images in one domain (in this case two 0s), and do a linear interpolation of the selected two images in the latent space. Then we translate all the interpolated images (including the two selected images) to all of the domains to generate "translated-interpolated" images—i.e., the corresponding interpolations in each of the domains.

As shown in Fig. 8, all "translated-interpolated" results can preserve the trend of the stroke width of the digits from the original interpolated domain. These results suggest that our approach aligns the domains so that some latent space directions have similar semantic meaning for all domains.

## F TABULAR/STRUCTURED DATA EXPERIMENT

We follow the same preprocessing for four UCI tabular datasets (Dua and Graff, 2017) from the MAF paper (Papamakarios et al., 2017), specifically MINIBOONE, GAS, HEPMASS, and POWER. In order to obtain a binary label for creating domains, we choose the last input feature for each dataset and discretize it based on whether it is higher or lower than the median value, which ensures the datasets are of equal size. We use the MLE version of AlignFlow as a baseline to compare with our model and use the AUB score as the metric for comparison.

As shown in the Table. 3, our method shows better performance with a large gap across all comparisons. This provides strong evidence that our proposed method (trained with learnable $Q$) better aligns the two domains than AlignFlow (trained with fixed $Q$).

It is worth emphasizing that there is no natural metric for evaluating GAN (Goodfellow et al., 2014) in tabular dataset. We believe this demonstrates one of the key benefits of our proposed method over GAN-based alignment methods.

## G MULTI-TRANSLATION TASKS IN CELEBA DATASET

In order to verify our proposed method performs well in a more complex setup than MNIST, we conducted experiments on the CelebA (Liu et al., 2015) dataset. We first create three domains (Black Hair, Blond Hair and Brown Hair) by using the hair color attribute. We further center crop

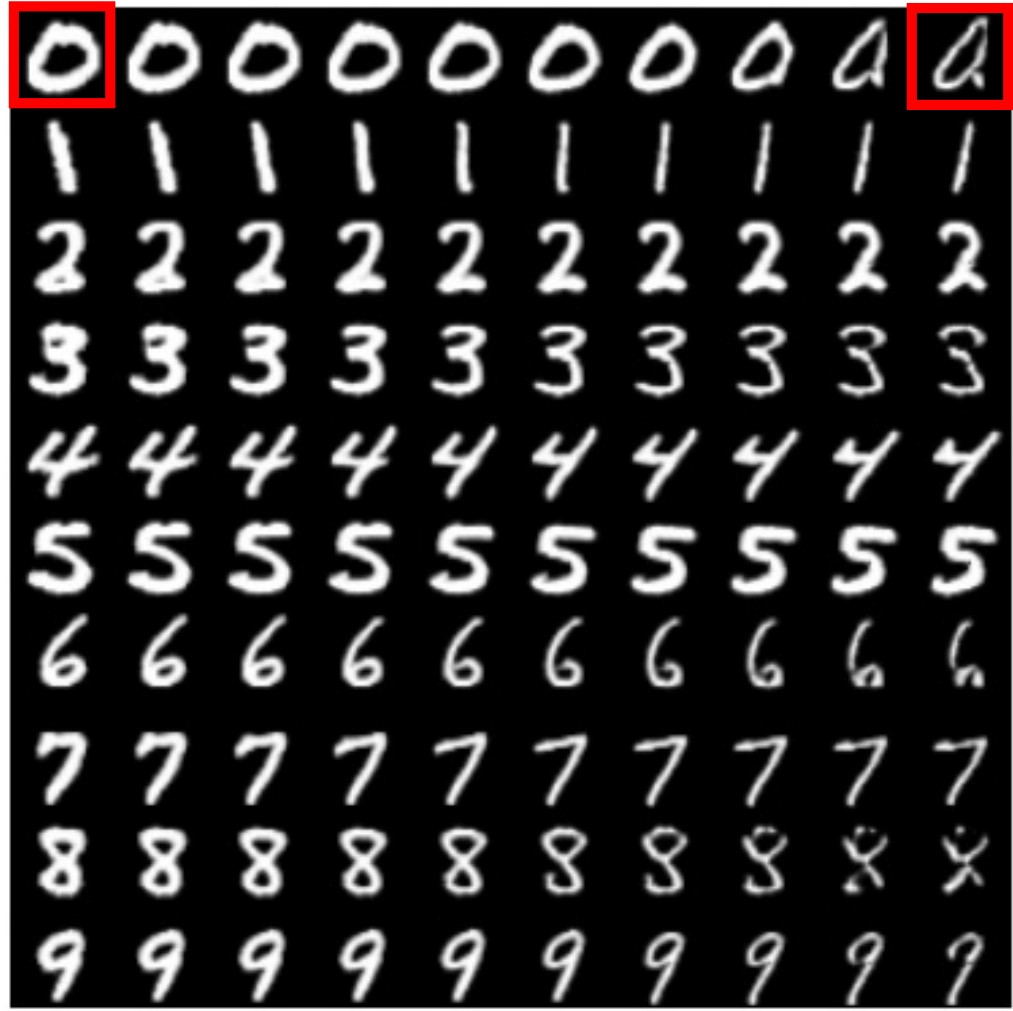

Figure 8: This figure shows the translation results of interpolated images. In the first row, the two images selected by red rectangles are real images from the dataset; and all eight images in between are generated by linear interpolation in the latent space. Starting from the second row, each row contains translated images which are transformed from the same latent vector in the same column and the first row.

Table 3: AUB score for the domain alignment task in four tabular datasets. AUB score is shown in nats (the lower the better).

|  | MINIBOONE | GAS | HEAPMASS | POWER |
|---|---|---|---|---|
| AlignFlow (MLE) | 13.26 | -5.65 | 19.57 | -0.89 |
| Ours | **-47.39** | **-137.91** | **-148.11** | **-91.17** |

the images to have the size $148 \times 148$ then resize them to $64 \times 64$. For the baseline model, we again uses MLE version of AlignFlow. We use both FID and AUB as a metric to compare the performances.

As shown in the Table 4, our proposed method demonstrates better performance in almost all experiments in terms of both FID and AUB. This consistently validates that the flexibility of $Q$ in our framework has clear advantages over the fixed $Q$ in AlignFlow for the domain alignment task. Intuitively, we think the tighter upper bound from our proposed method plays an important role in aligning distributions in the latent space.

We also provide qualitative results in Fig. 9. Compared to the AlignFlow (MLE) baseline, our model performs significantly better in translating between selected attributes. In particular, while our approach maintains a valid face structure during translation, AlignFlow is unable to maintain a valid face structure. Note that absolute image quality is not the focus of our paper as we use relatively simple flow models, but rather the comparison to prior methods and to show the feasibility of our approach in higher dimensions. We expect better qualitative results could be achieved using more advanced flow models such as GLOW, Residual Flows, Flow++, etc. but given that the scope of our work which focuses on the theoretical and foundational approach, we leave this to future work.

Table 4: This table provides evidence that our model has overall better performances than all baselines models in terms of both metrics. FID and AUB score for three images translation tasks in CelebA (for both metrics, lower is better). For domain names, 0 means Black Hair, 1 means Blond Hair, and 2 means Brown Hair. FID score for each translation task is calculated by averaging scores from each direction and AUB score is shown in nats.

|  | FID | | | | AUB | | | |
|---|---|---|---|---|---|---|---|---|
|  | $0\leftrightarrow1$ | $0\leftrightarrow2$ | $1\leftrightarrow2$ | Avg. | $0\leftrightarrow1$ | $0\leftrightarrow2$ | $1\leftrightarrow2$ | Avg. |
| AlignFlow (MLE) | 243.27 | **99.11** | 127.10 | 156.49 | **-38109.25** | -26437.60 | -26288.98 | -30278.61 |
| Ours, $\lambda = 0.01$ | **101.20** | 104.72 | **109.99** | **117.86** | -34041.39 | **-33884.92** | **-33902.84** | **-33943.05** |

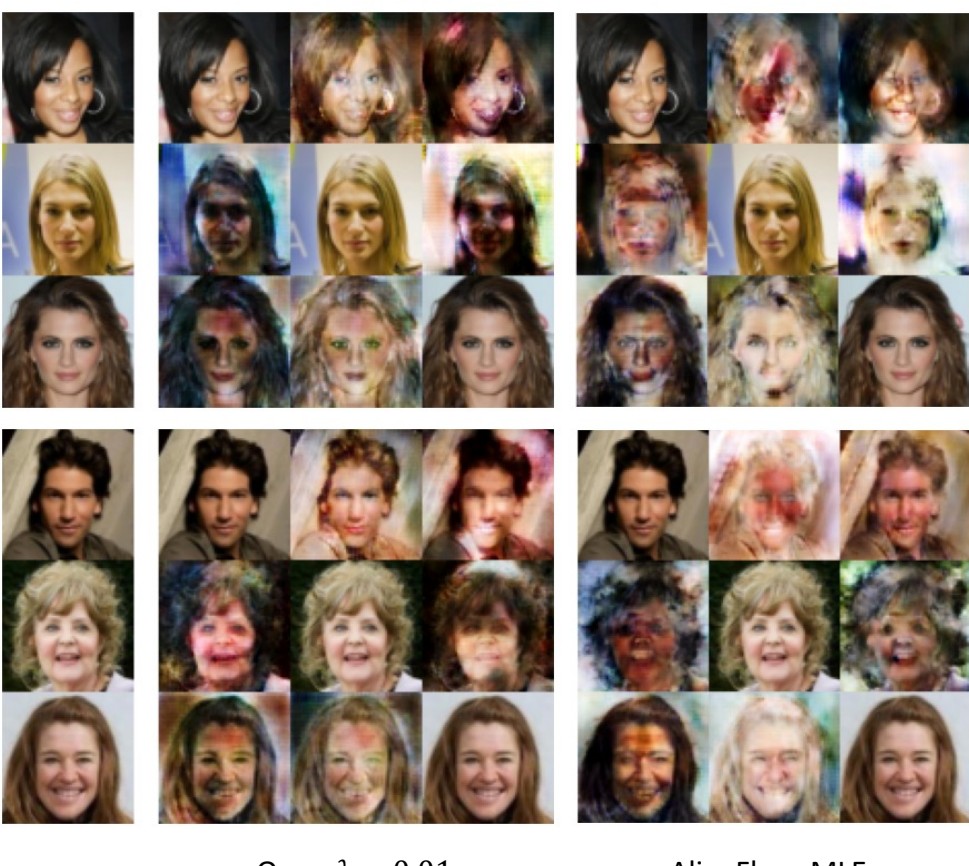

Ours, $\lambda = 0.01$        AlignFlow, MLE

Figure 9: This figure shows the translation results for three attributes (*Black Hair*, *Blonde Hair*, *Brown Hair*) in CelebA dataset. The first column corresponds to real images in the test dataset. The next three columns are translated results into three different attributes/domains (Each of the three columns represents translation to *Black Hair*, *Blonde Hair*, *Brown Hair* attribute respectively) from our model. The final three columns shows the translation results of MLE version of AlignFlow model with the same format as the previous three columns.

