# OpenReview forum: "Why be adversarial? Let's cooperate!: Cooperative Dataset Alignment via JSD Upper Bound"
_ICLR.cc/2022/Conference — ICLR 2022 Submitted_

### Official Review · Reviewer_fTCq · 2021-10-30

**Correctness:** 3
**Technical Novelty And Significance:** 3
**Empirical Novelty And Significance:** 3
**Recommendation:** 6
**Confidence:** 4

**Main Review:**

Strengths:
1. This paper theoretically gives an upper bound of GJSD with a learnable density function Q in a variational way, which extends prior method AlignFlow in multi-distribution alignment;
2. The authors give extensive discussions about the relationship between prior methods (including LRMF that tackles two data set alignment) and a regularization term.
3.  The author provides extensive experiments to illustrate its effectiveness. Moreover, the author also experimentally verify the statement in the discussions about prior methods and a regularization term.

Weaknesses:
1. Compared to AlignFlow, this work provides a learnable Q rather than a fixed function. This design seems reasonable. Could the authors give more discussions about the learnable Q? The authors state that a fixed normal distribution Q is insufficient. However, VAE[1] can generates very realistic figures from a normal distribution in a reversible way? Under what cases is a fixed Q not enough?

2. Compared to LRMF which tackles two data set alignment, this work can align multiple distributions. From Figure 3, the authors claim that if the density model class Q is not expressive enough, LRMF could fail to achieve the alignment. Could the authors explain the difference between the two methods (LRMF and yours) when optimizing the function Q?  Moreover, the number of the data sets to be aligned in the experiments is relatively small. When the number of distributions is large, is there some problems when the proposed method to align these distributions? This may be a common issue for existing methods, could the authors discuss it more?

[1]Kingma, Diederik P., and Max Welling. "Auto-encoding variational bayes." arXiv preprint arXiv:1312.6114 (2013)

**Summary Of The Paper:**

This paper proposes a flow-based method for the unsupervised data set alignment problem. It firstly reveals that the minimization problem over density models can be addressed by optimizing the upper bound of generalized JSD. Based on this theoretical result, the authors propose a new regularizer for the multi-distribution alignment problem. Compared to prior work, e.g., AlignFlow and LRMF, the authors derive a more general framework for the unsupervised data set alignment problem. The authors also provide extensive experiments to verify its effectiveness and superiority.

**Summary Of The Review:**

The framework advances the improvements in the problem of multi-distribution alignment, although it is an extension of prior works.

---

> ### Author Response · Authors · 2021-11-20
> **Response to Reviewer fTCq**
>
> ***Could the authors give more discussions about the learnable Q compared to AlignFlow? The authors state that a fixed normal distribution Q is insufficient. ... Under what cases is a fixed Q not enough?***
>
> We seek to answer this in three parts described below.
>
> 1) **Theoretical JSD gap**: Our approach with a learnable ${Q}$ can have a tighter bound than AlignFlow with a fixed ${Q}$ because the bound gap in Eq. 4 can be reduced during training as ${Q}$ is trained.
>
> 2) **Empirical usefulness**: As we illustrate in our toy experiments, if the flow models used for the $T_j$s do not have enough capacity or would get stuck in a local minimum (in practice, it is challenging to distinguish these two cases), the $Q$ model enables more flexibility.
> In particular, the $Q$ model could be ANY density model (i.e., it does not have to be a normalizing flow) including autoregressive models such as PixelCNN or even classical models such as Gaussian mixture models.
> Therefore, our learnable $Q$ is strictly a greater model class than AlignFlow (as we are able to use autoregressive density models like PixelCNN).
> In line with this, it is possible to use a learnable ${Q}$ that leverages a pretrained density model, which can be seen as similar to transfer learning.
> Thus, a learnable $Q$ can add additional modeling power and use transfer learning that is not possible with merely invertible $T_j$s.
>
> 3) **Shared parameters**: Especially for more than two distributions, sharing density estimation parameters across all distributions could be quite useful in terms of sample complexity and computational complexity compared to multiple independent flows as in AlignFlow.
>
> ***However, VAE[1] can generates very realistic figures from a normal distribution in a reversible way?***
>
> We are not quite sure what is meant here by referencing VAEs (though please correct or explain more if we have misunderstood).  VAEs are usually not invertible/reversible. They may be *approximately* invertible via the encoder/decoder structure. However, our flow-based models are *exactly* invertible.
> This exact invertibility is a key idea for flow-based models and key to our formulation.
> Without the invertibility of our $T_j$s, the AUB could not be computed.
> Also, we emphasize that our paper is not about the generative model task (in which a prior distribution is fixed to be a Gaussian), but rather the more general tasks of alignment and translation (in which both distributions are known only through samples).
> Moreover, while in theory, it is possible to transform any distribution to the normal distribution (e.g., using the optimal transport mapping), finding this map using only samples is the key challenge.
> And we show in our experiments that having a learnable $Q$ does have some empirical benefit.
> If we have misunderstood your comment about VAEs, please let us know as we hope we have not misunderstood things.
>
> [1] DP Kingma, M Welling (2013), Auto-encoding variational bayes, arXiv preprint arXiv:1312.6114
>
> ***Could the authors explain the difference between the two methods (LRMF and yours) when optimizing the function Q?***
>
> During training, the density function $Q$ in LRMF tries to model both the transformed data in the source domain and the data in the target domain, i.e. $Q$ models the data space; while our method tries to model the latent space generated by the two transformation functions from both domains, i.e. $Q$ models the latent space (Graphical illustration of different models can be visualized in Figure 2). In this way, our model is symmetric which naturally leads to multi-domain alignment structure, while LRMF model's asymmetric design does not.
>
>
> ***Moreover, the number of the data sets to be aligned in the experiments is relatively small. When the number of distributions is large, is there some problems when the proposed method to align these distributions? This may be a common issue for existing methods, could the authors discuss it more?***
>
> If the number of distributions indicates the number of domains, we may compare our Table 1 and Table 2 showing the increase of the number of domain reduces the relative performance gap in terms of both metrics (though our approach still performs better).
> We believe this is partly due to the limitation of the density model $Q$ because ten class distributions produces a significantly more complex shared latent space though we have not deeply investigated this particular concern.
> Indeed, we expect this to be a common challenge for any multi-distribution alignment method as the problem is naturally more complex.
> We expect that it would be possible to overcome some of this issue by learning a more complex $Q$ so that the AUB is a tighter bound to GJSD as illustrated in equation (4).

---

> > ### Author Response · Authors · 2021-11-27
> > **Have we answered your primary concerns?**
> >
> > Hi Reviewer fTCq
> >
> > Thank you for your thoughtful original review. In our response, we have added more discussion about our differences from AlignFlow and LRMF based on your comments.  We have also added experimental results on CelebA and structured data (see general response).
> >
> > Given our response, have we answered your primary concerns with the paper? If so, would you consider updating your score?
> >
> > Thank you for your time and consideration.

---

### Official Review · Reviewer_sZ2C · 2021-10-31

**Correctness:** 3
**Technical Novelty And Significance:** 2
**Empirical Novelty And Significance:** 4
**Recommendation:** 3
**Confidence:** 4

**Details Of Ethics Concerns:**

no.

**Main Review:**

## Concerns:

My major concern is why adversarial learning is challenging for distribution alignment? Could you provide a concrete example like an experiment? Do the authors justify this only according to, "However, adversarial learning can be quite challenging in practice", with two references are both earlier than 2019. How about nowadays? As the author claimed, "the auxiliary density model $Q_z$ may be more difficult to train than the auxiliary discriminator $D$". Does this seem to be contradictory to your justification?

1. The experiment design is trivial with MNIST. Using FID to measure generate data quality on MNIST is not well acknowledged. The authors should either choose a different dataset or report some other metrics.

2. There exist lots of multi-domain translation models with GANs, e.g. MUNIT and StarGAN v2. Without this kind of comparison, adversarial learning models shouldn't be criticized.

3. Since the proposed model introduces flow-based models, some ablation studies by changing the based model, e.g. Glow, NICE etc, would be better.

4. Since the flow-based model is used, is there any way to visualize the latent space on real data?

**Summary Of The Paper:**

The paper provides an interesting idea by unifying flow-based models under a non-adversarial framework to realize multiple-distribution alignment.

**Summary Of The Review:**

It's an interesting idea of using non-adversarial learning to realize multi-distribution alignment. But lots of concerns remain as given above. Thus, the impact of this paper is weak.

---

> ### Author Response · Authors · 2021-11-20
> **Response to Reviewer sZ2C (Part 1)**
>
> Thank you for your feedback. We hope to answer each of your questions and concerns below.
>
> ***My major concern is why adversarial learning is challenging for distribution alignment? Could you provide a concrete example like an experiment? Do the authors justify this only according to, "However, adversarial learning can be quite challenging in practice", with two references are both earlier than 2019. How about nowadays?***
>
> Unless we are mistaken (and please correct us if we have misunderstood), this concern seems to be primarily about the motivation for our work.
> We try to answer this concern in two distinct ways: 1) Clarifying that our approach is an alternative and not a replacement and 2) validating that overcoming challenges in adversarial learning is still an active area of research by giving several more recent references.
>
> 1) **Alternative and not replacement**: We realize that our language in our motivation might have communicated undue criticism of adversarial learning---which indeed has seen incredible empirical success in computer vision applications.
> Instead, we would like to communicate that our approach is an **alternative** rather than a replacement of adversarial learning (and we will adjust our language accordingly).
> We have modified our introduction and several other parts to reduce this harsh language (e.g., "Therefore, to avoid challenging adversarial learning" -> "Therefore, to provide an alternative to adversarial learning").
> Indeed, flow-based methods have different benefits and limitations compared to adversarial methods for distribution alignment.
> As one clear difference, our approach enables an application-agnostic yet theoretically grounded evaluation metric for comparing between models (even on tabular/structured datasets as detailed in our response to another reviewer).
> Additionally, our min-min problem is fundamentally different than a min-max problem and avoids issues unique to min-max problems (more details below) though our approach certainly has other limitations compared to GANs (e.g., it is restricted to invertible models and requires a density estimator).
> We do try to acknowledge these limitations in comparison to GANs in our original submission (see section 3).
> Finally, it should be possible to create a hybrid approach that may trade-off the strengths and weaknesses of both approaches depending on the application (as with AlignFlow).
> Thus, we ultimately want to convey our alignment approach is a feasible and fundamentally different alternative to adversarial, which is currently the only dominant approach.
> With our foundation, future work could focus on the performance aspects just as adversarial learning has been improving over the past seven years but is still an active area of research (see details below).
>
> 2) **Current challenges in adversarial learning**:
> To validate our claim that adversarial approaches are still challenging even today, we present a non-exhaustive list of recent papers that are still investigating these challenges (references below). While each of these papers propose solutions, they demonstrate that stability in GANs is still a critical unsolved problem and an active area of research.
>
>     - [1] This 2020 paper explores the theoretical conditions needed for stability and shows that existing GAN variants satisfy only some of these required conditions. The paper proposes a new more stable method for GAN training.
>     - [2] This 2020 paper explains that the min-max problem of standard GANs may not even have a Nash equilibrium.  They propose a new proximal equilibrium notion and a new adversarial training approach.
>     - [3] This 2020 paper mentions that adversarial training is unstable and proposes several mitigation techniques to stabilize the training.
>     - [4] This 2020 paper develops a theoretical viewpoint of min-max training and that suggests new learning rate schedules.
>     - [5] This 2020 paper analyze the local convergence behavior and propose a special Jacobian regularization to stabilize training.
>     - [6] This 2021 paper deconstructs the usual theoretical analysis of GAN optimization (particularly the assumption of discriminator optimality in most theoretical analysis) and shows that a new training regularization that improves the training algorithm.
>     - [7] This 2020 paper shows that standard non-saturating GAN training (ones that use the stabilizing "trick" from the original GAN paper) does in fact have a theoretical justification of minimizing an f-divergence.
>
> [1] Chu, C., Minami, K., & Fukumizu, K. (2020). Smoothness and Stability in GANs. In International Conference on Learning Representations.
>
> [2] Farnia, F., & Ozdaglar, A. (2020). Do GANs always have Nash equilibria?. In International Conference on Machine Learning (pp. 3029-3039). PMLR.

---

> > ### Author Response · Authors · 2021-11-20
> > **Response to Reviewer sZ2C (Part 2)**
> >
> > [3] Wu, Y., Zhou, P., Wilson, A. G., Xing, E., & Hu, Z. (2020). Improving GAN Training with Probability Ratio Clipping and Sample Reweighting. Advances in Neural Information Processing Systems, 33.
> >
> > [4] Cao, H., & Guo, X. (2020). Approximation and convergence of GANs training: an SDE approach. arXiv preprint arXiv:2006.02047.
> >
> > [5] Nie, W., & Patel, A. B. (2020, August). Towards a better understanding and regularization of GAN training dynamics. In Uncertainty in Artificial Intelligence (pp. 281-291). PMLR.
> >
> > [6] Munk, A., Harvey, W., & Wood, F. (2021, July). Assisting the Adversary to Improve GAN Training. In 2021 International Joint Conference on Neural Networks (IJCNN) (pp. 1-8). IEEE.
> >
> > [7] Shannon, M., Poole, B., Mariooryad, S., Bagby, T., Battenberg, E., Kao, D., ... & Skerry-Ryan, R. J. (2020). Non-saturating GAN training as divergence minimization. arXiv preprint arXiv:2010.08029.
> >
> >
> > ***As the author claimed, "the auxiliary density model  may be more difficult to train than the auxiliary discriminator ". Does this seem to be contradictory to your justification?***
> >
> > Thank you for bringing up this possible confusion.
> > First, we would like to clarify that while the *inner* optimization problem (i.e., density estimation in ours v.s. classification in GAN) may be more challenging in our approach compared to adversarial approaches, the global min-min problem may be more stable compared to a min-max problem.
> > Specifically, as we point out theoretically (Theorem 3), our min-min problem can align distributions even if $Q$ is from a restricted class of distributions and even if it is not optimal at every step.
> > Thus, we hope this clarifies the apparent contradiction and will clarify this in the final paper.
> > Second, as mentioned above, we would like to soften the justification to our approach as being an alternative rather than a replacement for adversarial learning.
> > Thus, while this is a potential challenge for our approach (which we acknowledge), we hope to alleviate the primary concern and confusion about our motivation and approach.
> >
> >
> > ***1. The experiment design is trivial with MNIST. Using FID to measure generate data quality on MNIST is not well acknowledged. The authors should either choose a different dataset or report some other metrics.***
> >
> > We would like to answer this in two parts: 1) new results on CelebA and 2) limitation of GAN-based metrics.
> >
> > 1) **New results on CelebA**:  To demonstrate our model’s performance on a more challenging dataset,  we conducted additional experiments on high-dimensional (64x64, CelebA [1]) dataset. We center-cropped CelebA dataset and separated domains based on hair color attribute, i.e., BlackHair, Blonde Hair, and Brown Hair.  As illustrated in Appendix G, we outperformed the baseline models in both AUB and FID metrics with an even bigger gap than in the MNIST experiment setup.
> > Please note that for the purpose of fast result, we use RealNVP as our invertible models since it is relatively easy and fast to train. The generated quality of the image is therefore restricted by the power of RealNVP (Please refer to the generated CelebA images in the original RealNVP paper). We expect better qualitative results could be achieved using more advanced flow models such as GLOW, Residual Flows, etc. but given the scope of our work which focuses on the theoretical and foundational approach, we leave this to future work.
> >
> > 2) **Limitation of GAN-based metrics**: We would like to note that "Using FID to measure generate data quality on MNIST is not well acknowledged." is *exactly* one of the problems we are trying to alleviate.  Theoretically grounded dataset-agnostic metrics do not exist for GAN-based models.
> > Our paper argues that this is indeed an unsolved problem for distribution alignment via adversarial methods.
> > Our AUB score provides a theoretically grounded metric that is dataset and domain agnostic and does not require any pretrained classifier.
> > It can be used on any dataset for comparing between models even structured/tabular datasets that cannot even be qualitatively validated (see new tabular results in response to another reviewer).
> > While we do not claim that our metric solves the alignment evaluation problem, we hope that our paper takes one step in the right direction, especially for tabular data (or even MNIST as you suggest).

---

> > > ### Author Response · Authors · 2021-11-20
> > > **Response to Reviewer sZ2C (Part 3)**
> > >
> > > ***2. There exist lots of multi-domain translation models with GANs, e.g. MUNIT and StarGAN v2. Without this kind of comparison, adversarial learning models shouldn't be criticized. 3. Since the proposed model introduces flow-based models, some ablation studies by changing the based model, e.g. Glow, NICE etc, would be better.***
> > >
> > > We emphasize that the goal of our paper is not to achieve state-of-the-art results on any particular dataset, task, or metric nor did we intend to unduly criticize GAN models.
> > > Rather, we are focused on developing a new theoretically grounded tool for distribution alignment that has fundamentally different strengths and weaknesses compared to adversarial learning.
> > > In particular, our paper focuses on comparing SOTA flow-based learning objectives and algorithms rather than comparing different model architectures.
> > > We do not claim any novelty in terms of model architecture and argue that our framework could be straightforwardly applied to more complex flow models (e.g., GLOW, Residual Flows, FFJORD, etc.) and density models (e.g., PixelCNN, VAEs, etc.).
> > > Given that we have limited GPU resources and our goal, we decided to focus on a simple fixed model setup but comparing the relative performance of different training objectives.
> > > If our paper is accepted, we will work to add additional flow-based models for the final version.
> > > Regarding direct comparison to GAN-based models, we do not directly compare to GAN-based models because we cannot compute our primary metric (i.e., AUB) for GAN-based models.
> > > Indeed, one of our core motivations (as in the LRMF paper) is that GAN-based methods do not have a theoretically grounded domain-agnostic metric for comparison.
> > > Finally, we note that a vast majority of normalizing flow literature (e.g., GLOW, Flow++, Residual Flow, etc.) does not compare directly with GAN-based models as flow-based and GAN-based approaches are fundamentally different and are challenging to compare on fair grounds (e.g., How do you know if GAN models have mode collapse? What about latent representations and interpolations in the real data space?).
> > >
> > > ***4. Since the flow-based model is used, is there any way to visualize the latent space on real data?***
> > >
> > > Excellent suggestion!  We have added the experiments in appendix E in which we illustrate the multi-domain translation results by interpolating between two real images in the latent space and showing the resulting images in different domains. We first randomly choose two real images in the same domain and get interpolated latent representations. Then we translate these interpolated latent representations to all domains to get the translated-interpolated images (Please refer to appendix section E for more details). As discussed in the appendix, we can clearly observe a consistent style changing pattern exists in all domains. We therefore argue that our model aligns different domains so that certain directions in the latent space have similar semantic meaning for all domains.

---

> > > > ### Author Response · Authors · 2021-11-27
> > > > **Have we answered your primary concerns?**
> > > >
> > > > Hi Reviewer sZ2C,
> > > >
> > > > Thank you for your thoughtful original review. We have improved the exposition and experimental results of the paper based on your comments including clarifying our motivation, adding experimental results on CelebA and structured data, and adding multi-domain interpolation results visualizing the latent space.
> > > >
> > > > Given our response, have we answered your primary concerns with the paper?
> > > >
> > > > Thank you for your time and consideration.

---

### Official Review · Reviewer_B4sF · 2021-11-03

**Correctness:** 3
**Technical Novelty And Significance:** 3
**Empirical Novelty And Significance:** 3
**Recommendation:** 6
**Confidence:** 4

**Main Review:**

* The major contribution of this paper is providing new domain alignment loss with theoretical guarantee and connections with JSD.

* The paper is well written and easy to follow. Details of experiments can be easily located and confirmed.

* A major concern of this paper is to which extend the current model can be applied to more general learning tasks. The current experiments (most of them) show the produced samples, compared with two SOTAs. It seems to me that the advantage is not very significant. See Fig. 6. Also, the quantitative results through FID and AUB are comparable to AlignFlow.

* Authors may be interested in showing more results of different learning tasks. As the approach is mainly for domain alignment, it would be interesting to show, e.g., visual domain adaptation results.

* More exploration and discussions are expected for high-dimensional or structured data, which may affect the design of transformation T and its inverse. The current model is only evaluated on simple or low-dimensional data, and the model generality is unclear.

* While GAN model is computationally expensive and hard to optimize, the proposed model did not prove itself to be efficient or more optimizable in conventional vision tasks where GAN was testified.


**Summary Of The Paper:**

The paper presents a unified framework for domain alignment through non-adversarial approaches. Instead of solving a min-max adversarial problem, this paper aims to solve a min-min optimization equivalent to minimizing the upper bound of JSD. It proves the equivalence and offers a computational and feasible solution that can be inserted into flow-based non-adversarial approaches in a plug-and-play fashion. Preliminary results on generating digits from different classes or domains have demonstrated the correctness of the model.

**Summary Of The Review:**

In brief, I was impressed by the theoretical results and connections of the new alignment loss and conventional JSD bound. Experiments seem to prove the model is workable on toy data and digits datasets. However, authors still need to explore and demonstrate its feasibility and value in more challenging learning tasks using high-dimensional or structured data.

---

> ### Author Response · Authors · 2021-11-20
> **Response to Reviewer B4sF (Part 1)**
>
> ***It seems to me that the advantage is not very significant. See Fig. 6. Also, the quantitative results through FID and AUB are comparable to AlignFlow.***
>
> We give a three-fold comment to this concern: 1) AlignFlow is special case, 2) our approach clearly outperforms AlignFlow in certain cases, and 3) new results on tabular data and CelebA.
>
> 1) **AlignFlow is a special case**: Our framework explicitly unifies the idea of AlignFlow as a special case. Indeed, our theoretical foundation that unifies prior methods is one of the key contributions. Thus, it is not surprising that in certain contexts the improvement may not be large.
>
> 2) **Our approach clearly outperforms AlignFlow in certain cases**: We believe it is clear in Table. 1 that our model shows better performance than the baselines in case of the small number of domains (e.g., the gaps between AlignFlow and ours in Table 1 are 18.98 (FID) and 29.8 (AUB) respectively, while the gaps in Table 2 are 5.76 (FID) and 36.37 (AUB)).
> While the improvements may be smaller for the 10 domains, we still believe they are significant.  Qualitatively in Figure 6, we believe there are several obvious qualitative differences; for example, row "1" translated to 2s and 3s, row "3" translated to 4s and 6s, row "4" translated to 5s,  row "8" translated to 5.
> In Table 2, while the gap is not as large, we still notice significant difference in terms of FID and better (though closer) AUB scores.
>
> 3) **New results on tabular data and CelebA**: Furthermore, to verify the superiority of our method over the baseline model, we additionally conducted comparisons on tabular dataset (two domains) and CelebA dataset (three domains). The specific descriptions on these experiments are provided below in the next answer. Overall, the results quantitatively demonstrate the superior performance of our model with a large gap. Hence, we argue that our approach outperforms the baseline approach on diverse datasets.
>
>
> ***Authors still need to explore and demonstrate its feasibility and value in more challenging learning tasks using high-dimensional or structured data.***
>
>
> In order to demonstrate our model's performance on a more challenging dataset, we conducted additional experiments on both high-dimensional (64x64, CelebA [1]) and structured data (UCI tabular datasets (POWER, GAS, HEPMASS, MINIBOONE)).
>
> We center-cropped CelebA dataset and separated domains based on hair color attribute, i.e., Black Hair, Blonde Hair, and Brown Hair. In case of the structured data, we used the data preprocessed by MAF [2]. In order for obtaining a binary label for the domain separation, we chose the last input feature for each dataset and discretized it based on its median.
>
> As shown in Fig. 9 and Table. 4 in appendix, our model demonstrates the better performance in aligning distributions than AlignFlow from different domains even in the higher dimensional case. Particularly in Fig. 9, the translated results of ours transforms the hair color decently while maintaining an original structure better than the baseline model did.
>
> Regarding the tabular dataset, across all the comparisons, our model shows the better performance in terms of AUB score over AlignFlow with a large gap. This verifies that the invertible functions trained with our flexible upperbound find the better alignment than the AlignFlow.
>
> [1] Liu, Ziwei and Luo, Ping and Wang, Xiaogang and Tang, Xiaoou (2015). Deep Learning Face Attributes in the Wild. ICCV
>
> [2] G Papamakarios, T Pavlakou, I Murray (2017). Masked Autoregressive Flow for Density Estimation. NeurIPS

---

> > ### Author Response · Authors · 2021-11-20
> > **Response to Reviewer B4sF (Part 2)**
> >
> > ***While GAN model is computationally expensive and hard to optimize, the proposed model did not prove itself to be efficient or more optimizable in conventional vision tasks where GAN was testified.***
> >
> >
> > We thank the reviewer for this suggestion.
> > We realize that our original submission may have communicated that our approach is inherently better in all aspects than adversarial learning.
> > Rather, we would like to emphasize that our approach should be viewed as an alternative rather than a replacement of GAN-based methods (see reply to reviewer sZ2C for more details).
> >
> > The main goal of this paper is to develop a new theoretically grounded tool for distribution alignment that has fundamentally different strengths and weaknesses compared to adversarial learning.
> > Theoretically, we prove that  our min-min problem can align distributions even if $Q$ is from a restricted class of distributions and even if it is not optimal at every step (Theorem 3).
> > This establishes some theoretical grounds for a simpler optimization problem though standard deep learning issues such as non-convexity and multiple local minima could remain.
> > From an empirical perspective, a thorough experimental comparison between methods in terms of efficiency or ease of optimization would be a great future direction but would likely require an entire paper as the methods are not easily compared.
> > For example, flows require invertible models but GANs cannot be evaluated with AUB. GANs can use arbitrary NNs but our approach can interpolate between real images via the latent space.
> > Or, how do you know if GANs have converged or if they have experienced mode collapse or overfitting?
> > Moreover, our model can be used beyond typical vision task as discussed in the previously where ours naturally has an evaluation metric while adversarial based models clearly lacks.

---

> > > ### Author Response · Authors · 2021-11-27
> > > **Have we answered your primary concerns?**
> > >
> > > Hi Reviewer B4sF,
> > >
> > > Thank you for your thoughtful original review. We have improved the exposition and experimental results of the paper based on your comments including adding experimental results on CelebA and structured data, and clarifying our paper's goal.
> > >
> > > Given our response, have we answered your primary concerns with the paper?  If so, would you consider updating your score?
> > >
> > > Thank you for your time and consideration.

---

### Author Response · Authors · 2021-11-20
**Response to all reviewers**

Thank you for all the reviewers for the valuable comments. We respond to each reviewer individually. Please check the revision of the paper as well. We marked the changes in blue color so that they are easy to notice. The revisions attempted to clarify our motivation as an alternative, rather than replacement of adversarial methods. We also add additional experiments on interpolation on the latent space, structured data, and high-dimensional data for tackling the concerns of reviewers. Please note that the experiment results are provided in Appendix E, F, G, respectively.

An overview of our additional quantitative results for structured/tabular and high-dimensional data is presented below.

**Structured/tabular data (AUB)**

|           | MINIBOONE | GAS     | HEPMASS | POWER  |
|-----------|-----------|---------|---------|--------|
| AlignFlow | 13.26     | -5.65   | 19.57   | -0.89  |
| Ours      | -47.39    | -137.91 | -148.11 | -91.17 |

**CelebA**

|           | FID (avg) | AUB (avg) |
|-----------|-----------|-----------|
| AlignFlow | 156.49    | -30278.61 |
| Ours      | 117.86    | -33943.05 |

---

### Decision · Program_Chairs · 2022-01-20

**Decision:**

Reject

**Comment:**

This paper offers flow-based alignment methods for alignment of distributions in a domain adaptation setting.  While there are many positive aspects of the submission, the experimental results only weakly support the results.  The AC agrees with the critical comments mentioned by reviewer sZ2C, and in particular observes that the experimentation is not state of the art with regard to current domain adaptation literature. Unfortunately the submission is not acceptable in present form.